# Community-based intervention for monitoring of salt intake in hypertensive patients: A cluster randomized controlled trial

**Pitchaporn Sonuch**[1], **Wichai Aekplakorn**[2], **Nophatee Pomsanthia**[3], **Natthida Boonyagarn**[3], **Siripak Makkawan**[4], **Suchada Thongchai**[4], **Wasinee Tosamran**[4], **Ananthaya Kunjang**[3], **Surasak Kantachuvesiri**[1,3]*

1 Division of Nephrology, Department of Medicine, Faculty of Medicine Ramathibodi Hospital, Mahidol University, Bangkok, Thailand, 2 Department of Community Medicine, Faculty of Medicine Ramathibodi Hospital, Mahidol University, Bangkok, Thailand, 3 Thai Low Salt Network, The Nephrology Society of Thailand, Bangkok, Thailand, 4 Office of Disease Prevention and Control 3, Nakhon Sawan, Thailand

* surasak.kan@mahidol.ac.th

**Data Availability Statement:** All relevant data are within the paper and its Supporting Information files

## Abstract

### Background

Excessive sodium intake is associated with high blood pressure and an increased risk of cardiovascular disease. However, reducing dietary salt has been challenging due to a lack of awareness and a high threshold in detecting saltiness.

### Objectives

The goal of this study is to evaluate the effectiveness of a combined intervention (intensive dietary education, food reformulation, environmental changes to facilitate salt reduction, and salt meter utilization), in comparison to standard education only, on salt intake and blood pressure.

### Methods

A cluster randomized-controlled trial was conducted on 219 hypertensive adults aged 18 to 70 years in Uthaithani, Thailand. Participants were randomized 1:1 into the intervention group (n = 111) and the control group (n = 108).

### Results

There were no differences in baseline characteristics between groups. The mean systolic and diastolic blood pressure was 143.6 and 82.1 mmHg and 142.2 and 81.4 mmHg in the intervention group, and the control group, respectively. The median 24-hour urinary sodium excretion was 3565 and 3312 mg/day, in the intervention and the control group, respectively. After 12 weeks, the change in systolic blood pressure was -13.5 versus -9.5 mmHg (P = 0.030) and diastolic blood pressure was -6.4 versus -4.8 mmHg (P = 0.164) in the intervention and control groups, respectively. Moreover, a reduction in 24-hour urine sodium excretion was observed [-575 versus -299 mg/day in the intervention and control groups,

**Funding:** This study was financially supported by the Thai Health Promotion Foundation No. 64-00255-0002 and World Health Organization (WHO) office, Thailand, registration 2021/1159185-0 The funders had no role in study design, data collection and analysis, decision to publish, or preparation of the manuscript.

**Competing interests:** The authors have declared that no competing interests exist.

respectively (P = 0.194)]. The change in 24-hour urine sodium excretion was statistically significant and reduced from baseline in the intervention group (P = 0.004). The dietary salt intake was significantly improved and was statistically different between groups (P = 0.035).

## Conclusions

The combined intervention significantly decreased systolic blood pressure and showed a trend towards reduced urine sodium excretion in hypertensive patients. These comprehensive approaches may be beneficial in reducing blood pressure and salt intake in the community.

## Clinical trial registration

This trial was registered at Clinicaltrials.gov with the identifier NCT05397054. https://classic.clinicaltrials.gov/ct2/show/NCT05397054

## Introduction

Non-communicable diseases (NCDs) are a global burden and a major cause of morbidity and mortality. NCDs are responsible for approximately 60% of all deaths and 43% of the global disease burden [1]. One of the most urgent public health issues related to NCDs is hypertension, which increases the risk of cardiovascular disease. High sodium intake which is the top three dietary risk factors was responsible for morbidity and mortality globally [2].

Excessive sodium intake is associated with high blood pressure [3, 4]. Consuming more dietary sodium than the World Health Organization's recommended daily amount of 2,000 milligrams is a risk factor for hypertension, cardiovascular disease, chronic kidney disease and is related to mortality [1]. According to the Prospective Urban Rural Epidemiology (PURE) study, 1-gram increment in urine sodium excretion was found to be correlated with an increase of 2.11 mmHg and 0.78 mmHg in systolic and diastolic blood pressure, respectively [5]. A reduction of dietary sodium intake can lead to lower blood pressure and lower cardiovascular morbidity and mortality [6–9].

According to a recent national survey in Thailand, dietary sodium consumption is 3,636 mg/day, which is nearly twice the recommended amount [10]. Most sodium consumption is derived from seasoning such as salt, fish sauce, soy sauce and seasoning powder. More than 60% of sodium is dissolved in the form of soup or curry. Therefore, reducing the use of seasoning, sauces, and soup can greatly reduce sodium consumption.

Dietary education and reformulation are effective methods for changing consumer behavior. However, these approaches are challenging for individuals to implement without adequate support or resources. One simple tool that can help individuals monitor their sodium intake is a salt meter, which measures the amount of sodium in food and beverages.

In addition, communities can play a vital role in reducing sodium consumption by promoting low-sodium options in restaurants, and implementing policies that reduce the sodium content of processed foods. By combining individual efforts with community-wide strategies, reducing the burden of sodium-related health problems can be achieved.

This study, was aimed to evaluate the effectiveness of comprehensive approaches that include intensive dietary education about risk of high sodium intake, dietary reformulation, environmental change in the community, and self-monitoring of salt intake by salt meter

compared to standard education alone in the community settings. The primary outcome is the change in 24-hour urine sodium excretion. Secondary outcomes include changes in blood pressure, as well as evaluations of changes in knowledge, attitudes, and behaviors after a complete 12-week follow-up.

## Materials and methods

### Study design

The study was a cluster randomized controlled trial. The intervention group was assigned to received intensive dietary education, dietary reformulation, environmental changes in community, and self-monitoring of salt intake by salt meter. The control group received standard education only. Participants were not blinded because the interventions could not be masked. The study protocol was approved by the Human Research Ethics Committee, Faculty of Medicine Ramathibodi Hospital, Mahidol University (COA. MURA2021/1004). This trial was registered at Clinicaltrials.gov with the identifier NCT05397054.

### Participants

The participants included in this study were aged between 18–70 years and had a diagnosis of hypertension (systolic blood pressure ≥130 or diastolic blood pressure ≥ 80 mmHg) [11]. Participants with end-stage kidney disease, pregnancy, or breastfeeding status, recent adjustment of any antihypertensive drugs or diuretics within 2 weeks prior enrollment and used salt supplements were excluded. During the conduction of the trial, participants were not allowed to adjust their antihypertensive or diuretic medication. If participants had systolic blood pressure exceeding 180 mmHg or presented with hypertensive emergencies, they were excluded from the study.

### Trial conduction

The trial was conducted in Uthaithani, Thailand from 10th of January to 13th of June 2022. Participants were recruited from twelve clusters (villages) in six healthcare centers, each serving a different community, including Muang Uthaithani, Nam Suem-Muang Uthaithani, Nong Chang, Khao Pathawi-Thap Than, Kokmo-Thap Than, and Paikeaw-Sawangarom. Each health center was responsible for 10–12 villages. We used a random number generator, to assign a random number (1, 2) to each village (cluster). The villages assigned the number 1 were the intervention group, and the villages assigned the number 2 were the control group. A total of 6 clusters were randomly allocated to the intervention group and another 6 clusters to the control group. A total of 20 patients meeting the eligible criteria were randomly recruited, using the random number generator, in each cluster. Informed consent was obtained from all participants. (Fig 1).

Baseline characteristics were collected, including age, sex, education, occupation, weight, height, waist circumference and medical history. Blood pressure was measured using an automatic blood pressure monitoring device (Omron HEM-7130-L) after 15 minutes of rest. Three measurements of blood pressure were recorded, and the average of both systolic and diastolic blood pressure were calculated. In addition, all participants were assessed using a standard questionnaire on their knowledge, attitude, and behavior, related to dietary consumption. The questionnaire was developed according to the local context and its content validity was assessed by 3 experts. It was tested for reliability, with the instrument achieving a Cronbach's alpha coefficient of 0.79 [12]. Participants were asked to collect 24-hour urine, starting with the second void, and including the first void on the next day.

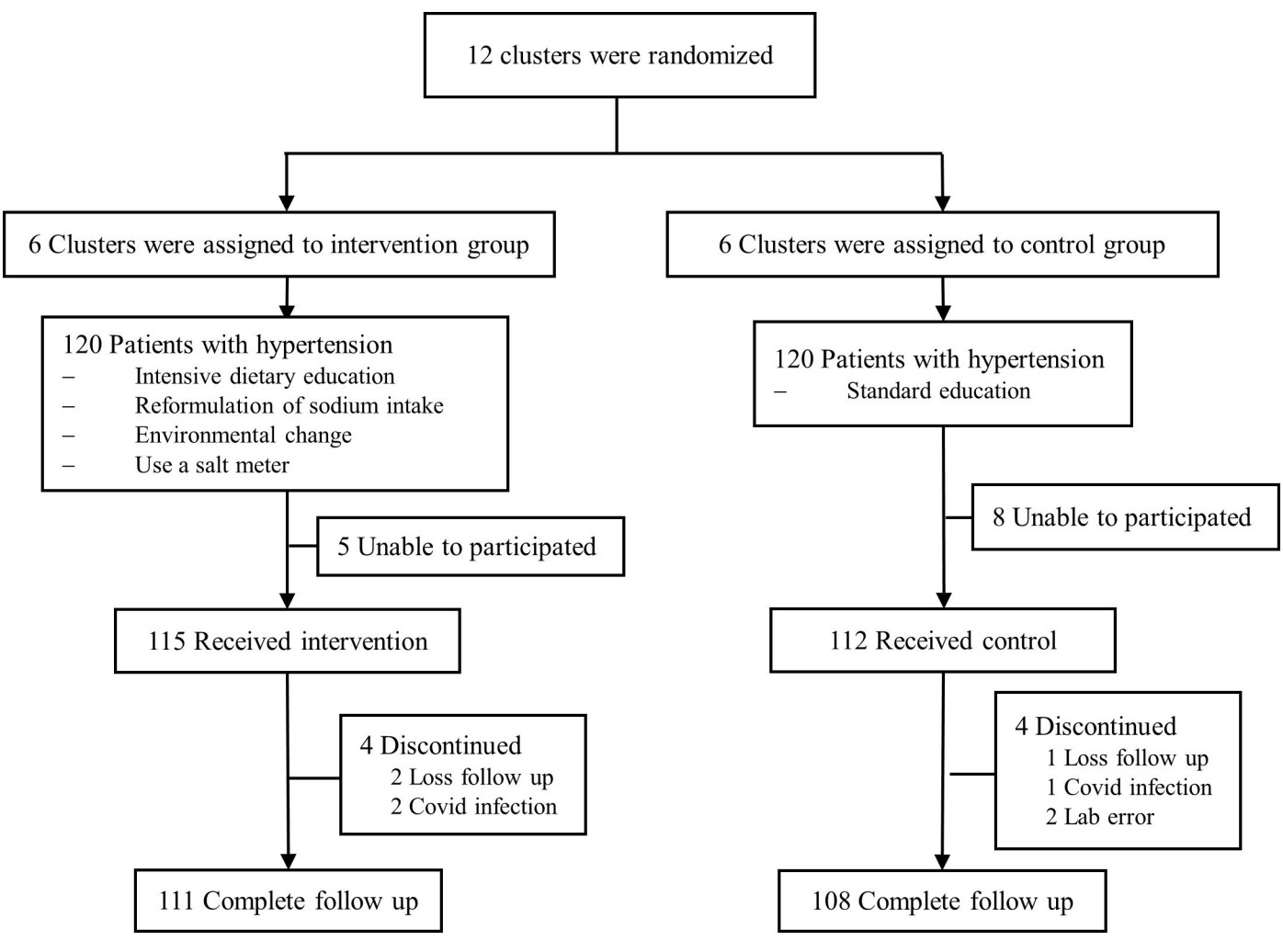

**Fig 1. Flow diagram of participants.**

In the intervention group, participants received four comprehensive interventions. First, intensive dietary education, participants received education from nurses and dietitians about high sodium intake, sources of sodium, and nutritional ingredients for 2 hours on day 0 and for 1 hour at 4 and 8 weeks. Second, dietary reformulation, they were taught to cook low-sodium recipes for home cooking. Third, environmental changes in the community, we educated and encouraged vendors to provide low sodium menus to community members. Fourth, using a salt meter, the participants were advised to utilize a salt meter provided to monitor and record the salt levels in their meals at least three times per week. The level of salt in food was reported as high, medium, and low. The salt meter used in this study was developed from Faculty of Engineering, Mahidol University, Thailand [13]. Participants in the control group received only standard education for blood pressure control. Blood pressure was evaluated in both groups every four weeks. The trial had a follow-up period of 12 weeks.

At 12 weeks, all participants were reevaluated regarding their knowledge, attitudes, and behaviors related to dietary intake using the same standard questionnaire. In addition, 24-hour urine samples were collected to evaluate urine sodium and creatinine excretion.

## Statistical analysis

Sample size was calculated with 90% statistical power and 5% alpha error, based on a previous study [14] to detect the mean change in blood pressure of 5 mmHg among intervention group and 0 mmHg among control group (standard deviation 5 mmHg). The design effect was calculated using the formula 1+(k-1)rho, where k represented the size of each cluster (20) and rho was the intracluster correlation (0.2). For cluster randomization, a sample size of 100 was required. Accounting for a 20% dropout rate, the final sample size of 120 participants per group was determined to provide sufficient power for detecting outcomes.

$$n = \frac{(a + b)^2 \left( SD_i^2 + SD_c^2 \right)}{(m_i - m_c)^2}$$

a = $Z_{alpha}$ = 1.96, b = $Z_{1\text{-beta}}$ = 1.28, SD = standard deviation of BP for intervention and control, mi-mc = mean changes of BP between intervention and control.

Baseline characteristics were evaluated and displayed in numbers with percentages, mean ± standard deviation (SD) or median with interquartile range (IQR). Categorical variables were analyzed using Fisher's exact test when expected values were less than 5, while continuous variables were analyzed using the student's t-test with 2-tailed test. The primary outcomes were evaluated using the Wilcoxon rank-sum test, while the Wilcoxon signed-rank test was used to compare urine sodium excretion within group between pre and post intervention and between two groups. We used an intention to treat analysis. Participants with incomplete data were excluded in the analysis. The change of blood pressure was performed with linear mixed-effects model (LMM) since the study measured blood pressure (BP) multiple times on each participant before and after intervention. LMMs account for the correlation between these repeated measures within each participant. We used an unstructured correlation structure.

T-test was used to compare the changes in knowledge, attitudes, and behaviors of sodium intake. In addition, the changes of variables between group and associated test of effect were estimated by LMM regression adjusted for clustering and baseline: age, sex, education, occupation, weight, height, body mass index and waist circumference with maximum likelihood estimation and accounting for missingness. Statistical significance was set at P value <0.05. All analyses were performed using STATA version 17.

## Results

240 participants were enrolled from 6 clusters/group and 219 completed follow-up (111 participants from six clusters and 108 participants from another six clusters in the intervention and the control group, respectively) (Fig 1). Demographic data and baseline characteristics are shown in Table 1. The mean age was approximately 60 years old. Most participants in the study were female as they were more willing to participate in the research than men. The mean systolic and diastolic blood pressure was 143.6 and 82.1 mmHg in the intervention group, while for the control group, the mean systolic and diastolic blood pressure was 142.2 and 81.4 mmHg, respectively. The median of 24-hour urinary sodium excretion was 3565 and 3312 mg/day, in the intervention and the control group, respectively. Fewer patients completed primary education, however, there were more patients with bachelor's degrees in the intervention group. In addition, the intervention group had a higher body weight, body mass index and waist circumference 1.

**Table 1. Demographic and baseline characteristics.**

| Baseline Characteristics | Intervention (111) | Control (108) | P value |
|---|---|---|---|
| Age, mean ± SD (years) | 60.0 ± 7.5 | 60.2 ± 7.5 | 0.883 |
| Sex | | | 0.121 |
| • Female (%) | 90 (81.1) | 78 (72.2) | |
| • Male (%) | 21 (18.9) | 30 (27.8) | |
| Education | | | 0.022* |
| • Uneducated (%) | 13 (11.7) | 10 (9.3) | 0.554 |
| • Primary school (%) | 62 (55.9) | 79 (73.2) | 0.008* |
| • Secondary school (%) | 26 (23.4) | 17 (15.7) | 0.152 |
| • Bachelor degrees (%) | 10 (9.0) | 2 (1.9) | 0.020* |
| Occupation | | | 0.818 |
| • Unemployed (%) | 32 (28.8) | 33 (30.6) | 0.780 |
| • Agriculturist, Employee (%) | 77 (69.4) | 72 (66.7) | 0.668 |
| • Government employee (%) | 2 (1.8) | 3 (2.8) | 0.680 |
| Weight, mean ± SD (kg) | 70.0 ± 14.1 | 64.6 ± 12.5 | 0.003* |
| Height, mean ± SD (cm) | 157.4 ± 7.5 | 156.4 ± 7.4 | 0.320 |
| BMI, mean ± SD (kg/m$^2$) | 28.2 ± 4.8 | 26.3 ± 4.4 | 0.004* |
| Waist circumference, mean ± SD (cm) | 94.1 ± 11.3 | 89.9 ± 9.9 | 0.004* |
| Blood pressure | | | |
| • Systolic, mean ± SD (mmHg) | 143.6 ± 13.4 | 142.2 ± 12.2 | 0.405 |
| • Diastolic, mean ± SD (mmHg) | 82.1 ± 9.7 | 81.4 ± 11.5 | 0.622 |
| Other underlying diseases | | | |
| • Diabetic mellitus (%) | 37 (33.3) | 28 (25.9) | 0.230 |
| • Dyslipidemia (%) | 71 (64.0) | 65 (60.2) | 0.564 |
| • Cardiovascular disease (%) | 5 (4.5) | 1 (0.9) | 0.213 |
| • Chronic kidney disease (%) | 6 (5.4) | 2 (1.9) | 0.280 |
| • Stroke (%) | 4 (3.6) | 1 (0.9) | 0.369 |
| • Chronic obstructive pulmonary disease (%) | 1 (0.9) | 1 (0.9) | 1.000 |
| 24 hours urine | | | |
| • Volume, median (IQR) (ml) | 2050 (1410–2910) | 1700 (1235–2555) | 0.047* |
| • Sodium excretion, median (IQR) (mmol/24h) | 155 (121–215) | 144 (107.5–191.5) | 0.096 |
| • Sodium excretion, median (IQR) (mg/24h) | 3565 (2783–4945) | 3312 (2473–4405) | 0.096 |
| • Na/Cr ratio, median (IQR) (mg/mg) | 3.6 (2.9–4.8) | 3.4 (2.6–4.5) | 0.346 |

*P value <0.05, statistical significance

## The changes in urinary sodium excretion and blood pressure

For the primary outcome, the difference in 24-hour urinary sodium excretion between groups was -276 mg/day, but this was not statistically significant (P = 0.194). The difference in the change of urinary sodium between two groups, adjusted for clustering and baseline variables, was not statistically different (P = 0.117) as shown in Table 2. However, 24-hour urinary sodium excretion was statistically significantly reduced from baseline (3565 mg/day) compared to at 12 weeks (3128 mg/day) in the intervention group (P = 0.004). Urinary sodium in the control group decreased slightly from baseline (3312 mg/day) to 12 weeks (3036 mg/day), but this change was not statistically significant (P = 0.267) (Fig 2).

There was a reduction in systolic blood pressure 13.5 ± 14.2 mmHg in the intervention group and 9.5 ± 12.8 mmHg in the control group at 12 weeks, making the difference between groups of -4.0 mmHg which was statistically significant (P = 0.030). This remained true even

**Table 2. The change of urine sodium excretion and blood pressure in both groups.**

| | Intervention | | | | Control | | | | Difference in changes between group P value$ | Difference in changes between group Adjusted P value# |
|---|---|---|---|---|---|---|---|---|---|---|
| | 0 week | 12 weeks | Δ change | P value Δ change | 0 week | 12 weeks | Δ change | P value Δ change | | |
| **Primary outcome** | | | | | | | | | | |
| Urine Na, median (IQR) (mg/24h) | 3565 (2783–4945) | 3128 (2208–4163) | -575 (-1541-552) | 0.004* | 3312 (2473–4405) | 3036 (2300–4600) | -299 (-1392-840) | 0.267 | 0.194 | 0.117 |
| **Secondary outcome** | | | | | | | | | | |
| SBP, mean ± SD (mmHg) | 143.6 ± 13.4 | 130.1 ± 13.4 | -13.5 ± 14.2 | <0.001* | 142.2 ± 12.2 | 132.7 ± 15.5 | -9.5 ± 12.8 | <0.001* | 0.030* | 0.021* |
| DBP, mean ± SD (mmHg) | 82.1 ± 9.7 | 75.7 ± 8.8 | -6.4 ± 8.7 | <0.001* | 81.4 ± 11.5 | 76.6 ± 9.8 | -4.8 ± 8.4 | <0.001* | 0.164 | 0.212 |

$P value different between group

#P value by regression adjusted for clustering and baseline: age, sex, education, occupation, weight, height, body mass index and waist circumference

*P value <0.05, statistical significance

after adjusting for clustering and baseline variables, p-value was 0.021. For the intra-group change, the systolic blood pressure significantly decreased from 143.6 ± 13.4 mmHg to 130.1 ± 13.4 mmHg in the intervention group (P<0.001) and from 142.2±12.2 mmHg to 132.7 ±15.5 mmHg in the control group (P<0.001). The diastolic blood pressure also significantly decreased from baseline of 82.1 ± 9.7 mmHg to 75.7 ± 8.8 mmHg at 12 weeks in the

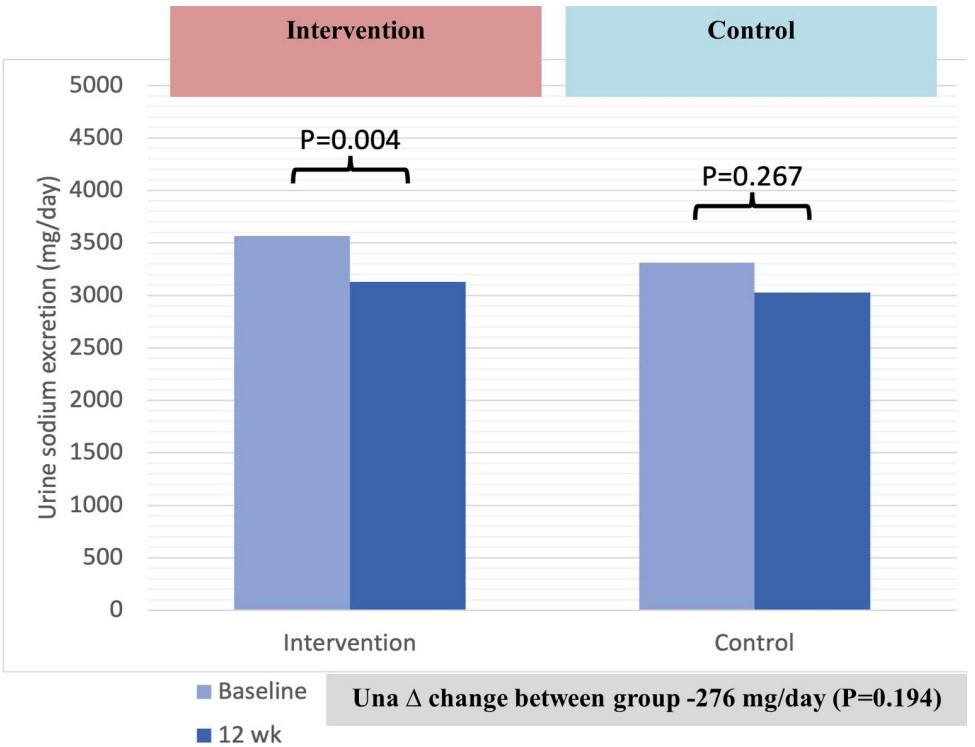

**Fig 2. The change of urine sodium excretion (Una).**

intervention group (P<0.001) and from 81.4 ± 11.5 mmHg to 76.6 ± 9.8 mmHg in the control group (P<0.001) (Fig 3). The difference in diastolic blood pressure reduction from baseline between the intervention group and control group was -1.6 mmHg at 12 weeks, but the difference was not statistically significant (P = 0.164). The intra-cluster correlation coefficients were 0.754 and 0.826 for systolic blood pressure and diastolic blood pressure respectively. Since the baseline body weight and BMI were statistically higher in the interventional group, we studied

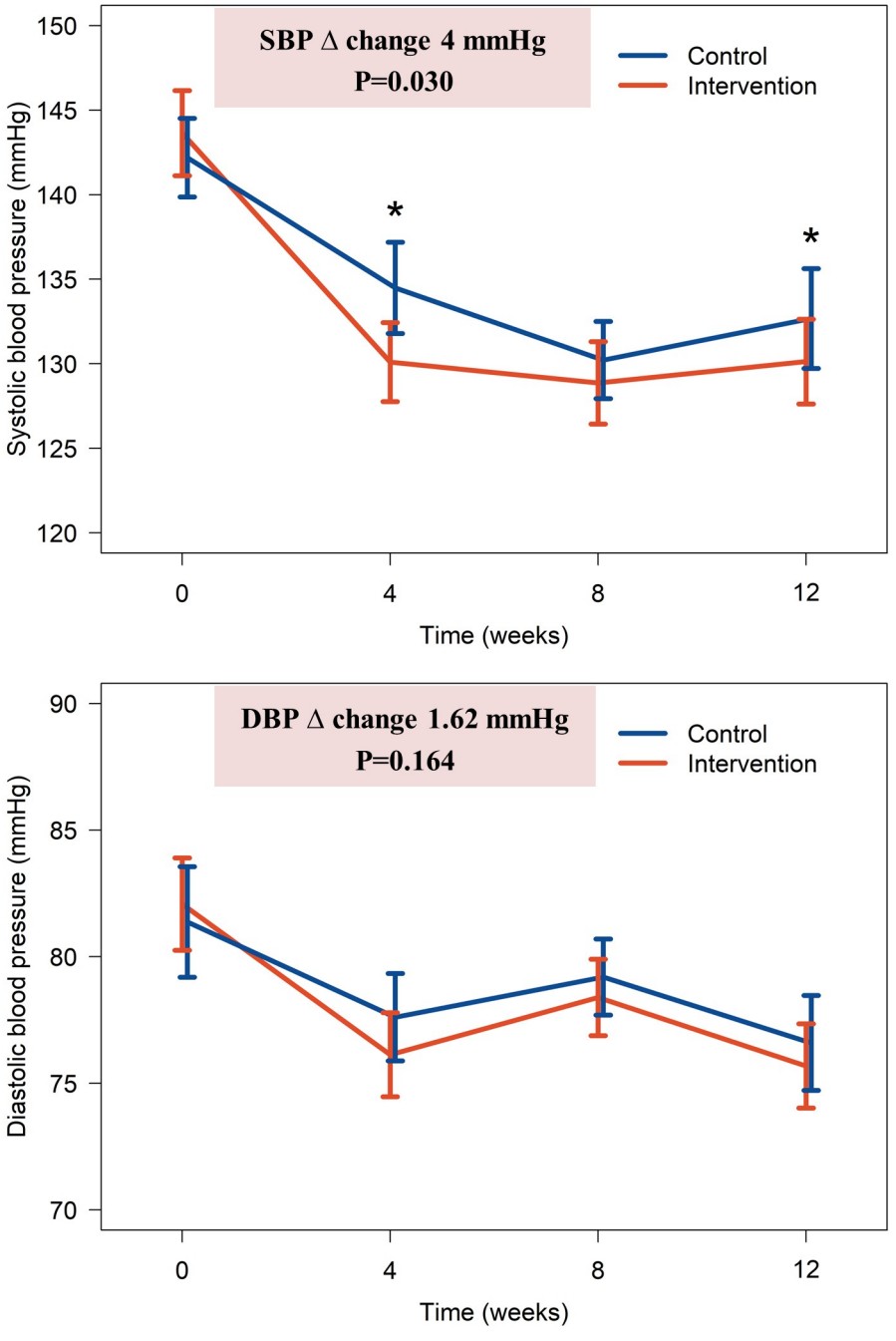

**Fig 3. The change of blood pressure.**

changes in these characteristics after 12 weeks of follow-up. There was no significant difference (P = 0.09) in the change in BMI from baseline between the two groups (BMI decreased by 0.03 and 0.23 kg/m$^2$ in the intervention and control groups, respectively). Therefore, the noticeable changes in blood pressure were not attributable to weight reduction.

### The changes in knowledge, attitudes, and behaviors about salt intake

The changes in knowledge, attitude, and behavior regarding salt consumption after 12 weeks of the experiment are presented in Table 3. The behavior score showed improvement in intervention group and was significantly different between groups after 12 weeks (P = 0.035), with a mean difference of 1.27. However, there were no significant differences in the knowledge score (P = 0.204) or attitude score (P = 0.460).

## Discussion

This study evaluated the effects of a comprehensive approach, which included intensive dietary education, food reformulation, community management and the use of a salt meter, on patients with hypertension. The results indicated significant reductions in systolic blood pressure due to these interventions, but no significant effect was observed on diastolic blood pressure. Urine sodium excretion had a trend to decrease significantly in the intervention group. These findings suggest that the comprehensive approaches for sodium reduction intervention were effective in reducing urine sodium excretion, as well as improving behavior related to sodium reduction. There was no significant improvement in knowledge about sodium reduction, which suggests that additional educational efforts may be needed to improve the understanding of the importance of sodium reduction.

Several studies have demonstrated effective sodium reduction strategies in reducing blood pressure and improving cardiovascular outcomes. Based on data from a systematic review, it was found that 21 recent interventions were successful in reducing dietary sodium. Interventions that emphasize individual education, either alone or in combination with tools to estimate sodium intake, have demonstrated positive outcomes in reducing salt consumption [14]. Most studies have mainly focused on education-based interventions. We observed the trend in reduction of blood pressure and 24-hour urinary sodium excretion in the control group during follow-up clinic visit, indicating that standard health education is properly implemented in our study. However, reducing dietary sodium intake has been challenging due to a lack of awareness and high threshold in detecting saltiness. Long-term consumption of high sodium

**Table 3. The changes in knowledge, attitudes, and behaviors about salt intake.**

| | Intervention | | | | Control | | | | Difference in changes between group P value$ | Difference in changes between group Adjusted P value# |
|---|---|---|---|---|---|---|---|---|---|---|
| | 0 week | 12 weeks | Δ change | P value Δ change | 0 week | 12 weeks | Δ change | P value Δ change | | |
| Knowledge mean ± SD (10) | 7.77 ± 1.74 | 8.14 ± 1.34 | 0.38 ± 2.08 | 0.058 | 7.83 ± 1.65 | 7.86 ± 1.72 | 0.03 ± 1.99 | 0.885 | 0.204 | 0.234 |
| Attitude mean ± SD (65) | 45.65 ± 9.73 | 47.95 ± 7.41 | 2.30 ± 10.14 | 0.019* | 44.20 ± 7.49 | 45.50 ± 6.56 | 1.30 ± 9.84 | 0.174 | 0.460 | 0.846 |
| Behavior mean ± SD (42) | 28.53 ± 4.75 | 30.59 ± 2.79 | 2.05 ± 5.21 | <0.001* | 29.09 ± 2.74 | 29.88 ± 2.45 | 0.79 ± 3.49 | 0.021* | 0.035* | 0.073 |

$P value different between group

#P value by regression adjusted for clustering and baseline: age, sex, education, occupation, weight, height, body mass index and waist circumference

*P value <0.05, statistical significance

is associated with decreased sensitivity to salty taste and increased threshold for detecting saltiness [15, 16], which is a significant contributing factor that leads to an increased tendency for individuals to consume higher amounts of sodium.

Compared to other studies, Morikawa et al. [17] investigated the effect of salt reduction on blood pressure using an electronic sensor and cellular phone. They found a significant decrease in systolic blood pressure after the intervention period. Takada et al. [18] also examined the effects of self-monitoring of salt intake using a simple electrical device. The study found that the intervention group had a significant reduction in salt consumption and a decrease in blood pressure. Another study by Yokokawa et al. [19] in Thailand focused on a dietary intervention in diabetic and hypertensive adults with high cardiovascular risk, resulting in significant reduction in systolic blood pressure, as well as improvements in lipid profiles and glycemic control. Finally, our previous study, the SMAL-SALT study [13] implemented a salt reduction intervention among hypertensive patients attending an outpatient clinic through a combination of dietary education and salt meter usage. We also found significant reductions in systolic blood pressure and daily sodium intake. However, their study did not assess changes in knowledge, attitudes, and behaviors related to sodium reduction. This is consistent with the findings of the current study that found a significant reduction in blood pressure and improved dietary habits in the intervention group.

Overall, these studies support the effectiveness of salt reduction strategies in reducing blood pressure, and suggest that strategies such as self-monitoring, nutritional education, and dietary counseling can be effective in promoting salt reduction. The findings from this study may be applied to a larger target group to help implement community salt reduction policies in Thailand. In Finland and the United Kingdom, the governments have launched strategies of salt awareness, in collaboration with the industry, and have mandated salt labeling. In Finland, these strategies were successful and lead to a significant decrease in salt intake from 14 gram/day in 1972 to 9 gram/day in 2002, resulting in a 10 mmHg decline in blood pressure and a 75–80% reduction in cardiovascular mortality [20]. Similarly, sodium reduction strategy in the United Kingdom, starting in 2003, resulted in a 17% decrease in salt intake between 2008/2009 and 2016/2017 [21–23].

The strengths of this study is the study design which is a cluster randomized control trial performed in the community to compare effects of the interventions on reducing sodium excretion and blood pressure. There are some limitations such as it was a non-blinded study due to interventions that were unable to be masked. The findings of knowledge, attitude, and behavior related to dietary consumption in this study can be considered as exploratory outcomes. We did not consider lifestyle changes (e.g. exercise) and baseline anti-hypertensive medications. This is due to the fact that in the rural primary healthcare setting where this study was done, there is a limited list of first-line medications for high blood pressure (thiazide diuretics, calcium channel blockers and angiotensin-converting enzyme inhibitors). Sodium-glucose cotransporters-2 inhibitors are not available in this setting. In addition, there were no reports of any symptoms related to hypertensive complications, nor any adjustment of medications in our patients. Furthermore, the results of this study may have limited generalizability as the local research team was selected from a group of talented and enthusiastic staff involved in the salt reduction campaign and the study was conducted in selected villages.

## Conclusions

The combined intervention of intensive dietary education, food reformulation, environmental changes in the community, and salt meter utilization significantly decreased systolic blood pressure and showed a trend towards reduced urine sodium excretion. These findings suggest

that implementing a comprehensive approach to sodium reduction in patients with hypertension may be effective in controlling salt intake and managing blood pressure in the community.

## Supporting information

**S1 Checklist. This is the CONSORT-2010-checklist of the project.**
(DOC)

**S1 Questionnaire. This is the questionnaire on the knowledge, attitude, and behavior, related to dietary consumption.** In Thai.
(PDF)

**S2 Questionnaire. This is the questionnaire on the knowledge, attitude, and behavior, related to dietary consumption.** In English.
(PDF)

**S1 Protocol. This is the protocol of study project in Thai.**
(PDF)

**S2 Protocol. This is the protocol of study project in English.**
(PDF)

**S1 Data. This is the data correction file.**
(XLSX)

## Acknowledgments

We would like to thank Mrs. Premthip Tiathaweekiat, Miss Nichanan Archtanyakam, Miss Nanara Padasittiphum, Mrs. Manawika Kulnee, Miss Apatsara Jantarat, Mrs. Kanokwan Pratheep, Mr. Tanakrit Junjajan, and Miss Kritaron Lao-in, the Uthaithani team for technical support and data collection.

## Author Contributions

**Conceptualization:** Wichai Aekplakorn, Siripak Makkawan, Surasak Kantachuvesiri.

**Data curation:** Pitchaporn Sonuch, Nophatee Pomsanthia, Natthida Boonyagarn.

**Formal analysis:** Wichai Aekplakorn.

**Funding acquisition:** Surasak Kantachuvesiri.

**Methodology:** Wichai Aekplakorn, Ananthaya Kunjang.

**Project administration:** Natthida Boonyagarn, Suchada Thongchai, Ananthaya Kunjang, Surasak Kantachuvesiri.

**Supervision:** Siripak Makkawan, Suchada Thongchai, Wasinee Tosamran.

**Visualization:** Wasinee Tosamran.

**Writing – original draft:** Pitchaporn Sonuch.

**Writing – review & editing:** Wichai Aekplakorn, Surasak Kantachuvesiri.

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
