## [Decision Letter · Decision Letter 0]

22 Apr 2024

PONE-D-24-07382Community-based intervention for monitoring of salt intake in hypertensive patients: A cluster randomized controlled trialPLOS ONE

Dear Dr. Kantachuvesiri,

Thank you for submitting your manuscript to PLOS ONE. After careful consideration, we feel that it has merit but does not fully meet PLOS ONE’s publication criteria as it currently stands. Therefore, we invite you to submit a revised version of the manuscript that addresses the points raised during the review process.

The manuscript has received an overall positive feedback, but would need refinement based on the reviewers suggestions. Kindly re-submit for further consideration.

We look forward to receiving your revised manuscript.

Kind regards,

Yogesh Kumar Jain, MPH

Academic Editor

PLOS ONE

Journal Requirements:

2. We note that you have selected “Clinical Trial” as your article type. PLOS ONE requires that all clinical trials are registered in an appropriate registry (the WHO list of approved registries is at      https://www.who.int/clinical-trials-registry-platform/network/primary-registries
https://www.who.int/clinical-trials-registry-platform/network/primary-registries and more information on trial registration is at http://www.icmje.org/about-icmje/faqs/clinical-trials-registration/).

Please state the name of the registry and the registration number (e.g. ISRCTN or ClinicalTrials.gov) in the submission data and on the title page of your manuscript.

a) Please provide the complete date range for participant recruitment and follow-up in the methods section of your manuscript.

b) If you have not yet registered your trial in an appropriate registry, we now require you to do so and will need confirmation of the trial registry number before we can pass your paper to the next stage of review. Please include in the Methods section of your paper your reasons for not registering this study before enrolment of participants started. Please confirm that all related trials are registered by stating: “The authors confirm that all ongoing and related trials for this drug/intervention are registered”.

Please see http://journals.plos.org/plosone/s/submission-guidelines#loc-clinical-trials for our policies on clinical trials.

"This study was financially supported by the Thai Health Promotion Foundation No. 64-00255-0002 and World Health Organization (WHO) office, Thailand, registration 2021/1159185-0 "

4. We note that the original protocol that you have uploaded as a Supporting Information file contains an institutional logo. As this logo is likely copyrighted, we ask that you please remove it from this file and upload an updated version upon resubmission.

Additional Editor Comments:

The manuscript has received an overall positive feedback, but would need refinement based on the reviewers suggestions. Kindly resubmit for further consideration.

Reviewers' comments:

Reviewer's Responses to Questions

**Comments to the Author**

1. Is the manuscript technically sound, and do the data support the conclusions?

Reviewer #1: Partly

Reviewer #2: Yes

2. Has the statistical analysis been performed appropriately and rigorously? 

Reviewer #1: No

Reviewer #2: Yes

3. Have the authors made all data underlying the findings in their manuscript fully available?

Reviewer #1: Yes

Reviewer #2: No

4. Is the manuscript presented in an intelligible fashion and written in standard English?

Reviewer #1: Yes

Reviewer #2: Yes

5. Review Comments to the Author

Reviewer #1: The manuscript is quite well written. However, the manuscript could be refined.

Comments

More information on the randomization method and process, and who performed it is to be provided.

Line 106, more information on the standard questionnaire on their knowledge, attitude, and behaviour is to be provided e.g. reliability and validation information of the questionnaire in the context of local setting.

For sample size calculation, the formula for the sample size and the value corresponding to each item is to be displayed.

Line 131-132, for Fisher’s exact test, 1 or 2-tailed test and Line 132, the specific name of the t-test to be stated.

Line 133, comparison between groups whether at certain time point(s) between the groups or the differences in pre and post-intervention between the two groups is to be stated.

Line 134, the statement ‘The change of blood pressure was performed with linear mixed-effects model’ and Line 135 with the statement ‘In addition, the changes of variables between the group and associated test of effect were estimated by regression’ requires more information. A statement on fulfilment of LMM assumptions is to be stated.

Line 134, as the change of blood pressure was measured at baseline, 4 weeks, 8 weeks and 12 weeks. Depending on the specific aim(s) of the study, the number of comparisons being made, the correlation structure of the data, and the balance between type I and type II errors, any adjustment to the p-value is to be clearly stated.

Line 136, the complete name of the regression test is to be stated.

The accepted level of statistical significance and whether 1 or 2 tailed p value is to be stated.

Intent to treat or per protocol analysis; missing data (if any) and method of handling missing data is to stated.

Table 1, Table 2 and Table 3, all statistical tests are to be denoted in the table footnote. For variables ‘other underlying diseases’ and ’24-hour urine’, the figures are to be placed in the same row with their respective category.

Table 2, effect size indices and 95% CI could be presented. The p-value in the change column is to be removed and placed in a separate column of each respective group. Presentation of p-value minus the symbol = e.g. P 0.004 is to be avoided. Likewise, with Table 3.

Ensure that all the information reported in the CONSORT checklist is presented or clearly presented in the manuscript.

Reviewer #2: Dear authors,

Congratulations for conducting and reporting the study. This study determines the effectiveness of a combined intervention on reducing salt intake and blood pressure relative to health education. The manuscript is well-written in standard English with sufficient data presentation, analysis, and discussion. The manuscript fulfills the stated aims and meets the standard for publication.

Strengths of the study

Authors have clearly established the research gaps and objectives. The chosen study design (cluster-randomised trial) is suitable to meet the study objectives. Data has been adequately presented and interpretated. Discussion is balanced and conclusion is based on the results and analysis.

Weaknesses of the study

Method:

Line 85: definition of hypertension is misleading. Current statement is not correct and citation is missing.

Line 91: Please provide details when clusters were randomised (e.g before or after the participants were recruited) and methods for randomisation of participants.

Line 87 and 88: The expression "During the conduction of the trial, participants were not allowed to adjust their antihypertensive or diuretic medication" is misleading. Is it ethical not to allow participants to alter the drug?

Line 105: Authors did not register the "knowledge, attitude, and behavior related to dietary consumption" related outcome in the trial registry. This can be considered as exploratory outcomes only.

Line 109 to 116: Please provide the details of intervention (e.g duration and sessions of health education), so that the intervention could be replicated.

Results:

Table 1: *is missing in the table. What indicates * for.

Table 2: Please report intra-cluster correlation coefficients for the reported outcomes. These are important for future studies to estimate power and meta-analyse.

Discussion:

Line 217: "Furthermore, we observed the trend in reduction of blood pressure and 24-hour urinary sodium excretion in the control group which may be due to patient education or unknown intervention during follow-up clinic visit". It is not the limitation of the study. Instead, This is the indication that standard health education is properly implemented. Therefore, this could be discussed separately.

Line 219: The generalisability is limited as the study was conducted in selected villages.

There are additional limitations. One limitation is not considering important lifestyle changes (e.g exercise). Authors did not report baseline data on history of current anti-hypertensive medication, which either could be adjusted in the analysis or acknowledged in the limitation.

It would be easier to follow the structure when strength and limitations are in the last sentence of discussion section.

Limitations:

Language:

There are a few errors in English language: e.g

Line 198: Long term is Long-term

Incorporating these comments will improve the quality of the manuscript.

6. PLOS authors have the option to publish the peer review history of their article (what does this mean?). If published, this will include your full peer review and any attached files.

Reviewer #1: No

Reviewer #2: **Yes: **Mahesh Kumar Khanal

---

## [Author Response · Author response to Decision Letter 0]

5 Jun 2024

Response to reviewers

Reviewer #1: The manuscript is quite well written. However, the manuscript could be refined.

Comments

More information on the randomization method and process, and who performed it is to be provided. – completed, lines 96–102 of the tracked changes version . 

Ans. Each health center was responsible for 10-12 villages. We used a random number generator, to assign a random number (1, 2) to each village (cluster). The villages assigned the number 1 were the intervention group, and the villages assigned the number 2 were the control group. A total of 6 clusters were randomly allocated to the intervention group and another 6 clusters to the control group. A total of 20 patients meeting the eligible criteria were randomly recruited, using the random number generator, in each cluster. Informed consent was obtained from all participants. (Fig 1)

Line 106, more information on the standard questionnaire on their knowledge, attitude, and behaviour is to be provided e.g. reliability and validation information of the questionnaire in the context of local setting. – completed, lines 109-112. 

Ans. The questionnaire was developed according to the local context and its content validity was assessed by 3 experts. It was tested for reliability, with the instrument achieving a Cronbach's alpha coefficient of 0.79 [12]. 

For sample size calculation, the formula for the sample size and the value corresponding to each item is to be displayed. – completed, lines 135-137. 

Ans.

n= ((a+b)^(2 ) (〖SD〗_i^2+〖SD〗_c^2 ))/((m_i-m_c )^2 )

a= Z alpha=1.96, b= Z1-beta=1.28, SD= standard deviation of BP for intervention and control, mi-mc =mean changes of BP between intervention and control.

Line 131-132, for Fisher’s exact test, 1 or 2-tailed test and Line 132, the specific name of the t-test to be stated. – completed, lines 139-141. 

Ans. Categorical variables were analyzed using Fisher's exact test when expected values were less than 5, while continuous variables were analyzed using the student’s t-test with 2-tailed test.

Line 133, comparison between groups whether at certain time point(s) between the groups or the differences in pre and post-intervention between the two groups is to be stated. – completed, lines 141-143 . 

 Ans. The primary outcomes were evaluated using the Wilcoxon rank-sum test, while the Wilcoxon signed-rank test was used to compare urine sodium excretion within group between pre and post intervention and between two groups. 

Line 134, the statement ‘The change of blood pressure was performed with linear mixed-effects model’ and Line 135 with the statement ‘In addition, the changes of variables between the group and associated test of effect were estimated by regression’ requires more information. A statement on fulfilment of LMM assumptions is to be stated. – completed, lines 144-146. 

 Ans. The change of blood pressure was performed with linear mixed-effects model (LMM) since the study measured blood pressure (BP) multiple times on each participant before and after intervention. LMMs account for the correlation between these repeated measures within each participant. We used an unstructured correlation structure. 

Line 134, as the change of blood pressure was measured at baseline, 4 weeks, 8 weeks and 12 weeks. Depending on the specific aim(s) of the study, the number of comparisons being made, the correlation structure of the data, and the balance between type I and type II errors, any adjustment to the p-value is to be clearly stated. – completed, line 146. 

Ans. We used an unstructured correlation structure. 

Line 136, the complete name of the regression test is to be stated. completed, line 144. 

Ans. The change of blood pressure was performed with linear mixed-effects model (LMM) since the study measured blood pressure (BP) multiple times on each participant before and after intervention.

The accepted level of statistical significance and whether 1 or 2 tailed p value is to be stated. -completed, lines 140-141 and 151-152. 

Ans. The continuous variables were analyzed using the student’s t-test with 2-tailed test. 

Statistical significance was set at P-value <0.05.

Intent to treat or per protocol analysis; missing data (if any) and method of handling missing data is to stated. -completed, line 143-144. 

Ans. We used an intention to treat analysis. Participants with incomplete data were excluded in the analysis.

Table 1, Table 2 and Table 3, all statistical tests are to be denoted in the table footnote. For variables ‘other underlying diseases’ and ’24-hour urine’, the figures are to be placed in the same row with their respective category. 

Ans. We added P value and the figures. All statistical tests are to be denoted in the table footnote in Table 1,2 and 3 as suggested

Table 2, effect size indices and 95% CI could be presented. The p-value in the change column is to be removed and placed in a separate column of each respective group. Presentation of p-value minus the symbol = e.g. P 0.004 is to be avoided. Likewise, with Table 3. -completed, in Table 1,2 and 3. 

Ensure that all the information reported in the CONSORT checklist is presented or clearly presented in the manuscript. -Completed as suggested in the manuscript.

Reviewer #2: Dear authors,

Congratulations for conducting and reporting the study. This study determines the effectiveness of a combined intervention on reducing salt intake and blood pressure relative to health education. The manuscript is well-written in standard English with sufficient data presentation, analysis, and discussion. The manuscript fulfills the stated aims and meets the standard for publication.

Strengths of the study

Authors have clearly established the research gaps and objectives. The chosen study design (cluster-randomised trial) is suitable to meet the study objectives. Data has been adequately presented and interpretated. Discussion is balanced and conclusion is based on the results and analysis.

Weaknesses of the study

Method:

Line 85: definition of hypertension is misleading. Current statement is not correct and citation is missing. -completed, line 85 of the tracked changes version. 

Ans. The participants included in this study were aged between 18-70 years and had a diagnosis of hypertension (systolic blood pressure ≥130 or diastolic blood pressure ≥ 80 mmHg)[11].

Line 91: Please provide details when clusters were randomised (e.g before or after the participants were recruited) and methods for randomisation of participants. -completed, line96-102. 

Ans. Each health center was responsible for 10-12 villages. We used a random number generator, to assign a random number (1, 2) to each village (cluster). The villages assigned the number 1 were the intervention group, and the villages assigned the number 2 were the control group. A total of 6 clusters were randomly allocated to the intervention group and another 6 clusters to the control group. A total of 20 patients meeting the eligible criteria were randomly recruited, using the random number generator, in each cluster. Informed consent was obtained from all participants. (Fig 1)

Line 87 and 88: The expression "During the conduction of the trial, participants were not allowed to adjust their antihypertensive or diuretic medication" is misleading. Is it ethical not to allow participants to alter the drug? -completed, lines 87-90.

Ans. If participants had systolic blood pressure exceeding 180 mmHg or presented with hypertensive emergencies, they were excluded from the study.

Line 105: Authors did not register the "knowledge, attitude, and behavior related to dietary consumption" related outcome in the trial registry. This can be considered as exploratory outcomes only.

Ans. We agree and added in the discussion lines 242-244.

Line 109 to 116: Please provide the details of intervention (e.g duration and sessions of health education), so that the intervention could be replicated. -completed, lines 115-116.

Ans. First, intensive dietary education, participants received education from nurses and dietitians about high sodium intake, sources of sodium, and nutritional ingredients for 2 hours on day 0 and for 1 hour at 4 and 8 weeks.

Results:

Table 1: *is missing in the table. What indicates * for.

Ans. We added * and P value in the table and all statistical tests are to be denoted in the table footnote in Table 1,2 and 3 as suggested

Table 2: Please report intra-cluster correlation coefficients for the reported outcomes. These are important for future studies to estimate power and meta-analyse. -completed, lines 189-190.

Ans. The intra-cluster correlation coefficients were 0.754 and 0.826 for systolic blood pressure and diastolic blood pressure respectively.

Discussion:

Line 217: "Furthermore, we observed the trend in reduction of blood pressure and 24-hour urinary sodium excretion in the control group which may be due to patient education or unknown intervention during follow-up clinic visit". It is not the limitation of the study. Instead, This is the indication that standard health education is properly implemented. Therefore, this could be discussed separately.

Ans. We agree and edited in the discussion as suggested on lines 212-214

Line 219: The generalisability is limited as the study was conducted in selected villages. -completed, line 248.

There are additional limitations. One limitation is not considering important lifestyle changes (e.g exercise). Authors did not report baseline data on history of current anti-hypertensive medication, which either could be adjusted in the analysis or acknowledged in the limitation. -completed, lines 244-246.

It would be easier to follow the structure when strength and limitations are in the last sentence of discussion section. 

Ans: We agree and edited in the discussion as suggested on lines 240-248.

Limitations:

Language:

There are a few errors in English language: e.g

Line 198: Long term is Long-term -completed, line 215.

Journal Requirements:

We have checked and ensured our manuscript meets PLOS ONE’s requirements.

2. We note that you have selected “Clinical Trial” as your article type. PLOS ONE requires that all clinical trials are registered in an appropriate registry (the WHO list of approved registries is at https://www.who.int/clinical-trials-registry-platform/network/primary-registries" https://www.who.int/clinical-trials-registry-platform/network/primary-registries and more information on trial registration is at http://www.icmje.org/about-icmje/faqs/clinical-trials-registration/).

Please state the name of the registry and the registration number (e.g. ISRCTN or ClinicalTrials.gov) in the submission data and on the title page of your manuscript. completed, line 15-16. 

a) Please provide the complete date range for participant recruitment and follow-up in the methods section of your manuscript. completed, line 93. 

b) If you have not yet registered your trial in an appropriate registry, we now require you to do so and will need confirmation of the trial registry number before we can pass your paper to the next stage of review. Please include in the Methods section of your paper your reasons for not registering this study before enrolment of participants started. Please confirm that all related trials are registered by stating: “The authors confirm that all ongoing and related trials for this drug/intervention are registered”.

Please see http://journals.plos.org/plosone/s/submission-guidelines#loc-clinical-trials for our policies on clinical trials.

"This study was financially supported by the Thai Health Promotion Foundation No. 64-00255-0002 and World Health Organization (WHO) office, Thailand, registration 2021/1159185-0 "

Please include this amended Role of Funder statement in your cover letter; we will change the online submission form on your behalf. -Completed in covering letter.

4. We note that the original protocol that you have uploaded as a Supporting Information file contains an institutional logo. As this logo is likely copyrighted, we ask that you please remove it from this file and upload an updated version upon resubmission. -Completed and new file attached

---

## [Decision Letter · Decision Letter 1]

21 Jun 2024

PONE-D-24-07382R1Community-based intervention for monitoring of salt intake in hypertensive patients: A cluster randomized controlled trialPLOS ONE

Dear Dr. Kantachuvesiri,

Thank you for submitting your manuscript to PLOS ONE. After careful consideration, we feel that it has merit but does not fully meet PLOS ONE’s publication criteria as it currently stands. Therefore, we invite you to submit a revised version of the manuscript that addresses the points raised during the review process.

We look forward to receiving your revised manuscript.

Kind regards,

Peng Gao, Ph.D.

Academic Editor

PLOS ONE

Journal Requirements:

Reviewers' comments:

Reviewer's Responses to Questions

**Comments to the Author**

1. If the authors have adequately addressed your comments raised in a previous round of review and you feel that this manuscript is now acceptable for publication, you may indicate that here to bypass the “Comments to the Author” section, enter your conflict of interest statement in the “Confidential to Editor” section, and submit your "Accept" recommendation.

Reviewer #1: All comments have been addressed

Reviewer #2: All comments have been addressed

Reviewer #3: All comments have been addressed

2. Is the manuscript technically sound, and do the data support the conclusions?

Reviewer #1: Partly

Reviewer #2: Yes

Reviewer #3: Partly

3. Has the statistical analysis been performed appropriately and rigorously? 

Reviewer #1: Yes

Reviewer #2: Yes

Reviewer #3: Yes

4. Have the authors made all data underlying the findings in their manuscript fully available?

Reviewer #1: Yes

Reviewer #2: Yes

Reviewer #3: Yes

5. Is the manuscript presented in an intelligible fashion and written in standard English?

Reviewer #1: Yes

Reviewer #2: Yes

Reviewer #3: Yes

6. Review Comments to the Author

Reviewer #1: (No Response)

Reviewer #2: Dear authors,

Thank you for addressing my previous comments.

I only have a few minor corrections. Please make the writing of p value consistent, either p or P. All tables contain the redundant letter P, column head with P value is enough. To improve reading, all the table titles should be just before each table. These corrections can be made even during production phase.

Congratulations!

Kind regards,

Reviewer #3: The authors have adequately addressed most of the comments raised in a previous round of review. This manuscript needs further improvement.

1.The positioning of the modifications made in the author's response does not match the actual situation, resulting in reading difficulties.

2. It would be preferable to use a three-line table format in the assay

3. The patients recruited exhibited a significant age range, with the youngest being only 18 years old. But the exclusion criteria do not include secondary hypertension, such as primary aldosteronism. Salt intake may play a more significant role in this condition.

4. The author provided detailed imformation about the randomization method and process. Line 207,“240 participants were enrolled from 6 clusters/group”, 12 clusters might be an accurate representation according to my understanding.

5.In the discussion section, the author noted that there was no adjustment of medication in patients. It is puzzling that the blood pressure in the control group still showed a significant decline.

6.As mentioned before, the presentation of p-value should not minus the symbol = e.g. P 0.004.

7. PLOS authors have the option to publish the peer review history of their article (what does this mean?). If published, this will include your full peer review and any attached files.

Reviewer #1: No

Reviewer #2: No

Reviewer #3: No

---

## [Author Response · Author response to Decision Letter 1]

15 Jul 2024

Reviewer #2: I only have a few minor corrections. Please make the writing of p value consistent, either p or P. All tables contain the redundant letter P, column head with P value is enough. To improve reading, all the table titles should be just before each table. These corrections can be made even during production phase.

Ans. We used P value consistently in the manuscript. We deleted P value in all the tables except column head and put the titles before each table as suggested. 

Reviewer #3: The authors have adequately addressed most of the comments raised in a previous round of review. This manuscript needs further improvement.

1. The positioning of the modifications made in the author's response does not match the actual situation, resulting in reading difficulties. 

Ans. We revised the manuscript which the positioning of modification in the author’s response to match the actual situation as suggested.

2. It would be preferable to use a three-line table format in the assay

Ans. We changed all tables to a three-line table format as suggested.

3. The patients recruited exhibited a significant age range, with the youngest being only 18 years old. But the exclusion criteria do not include secondary hypertension, such as primary aldosteronism. Salt intake may play a more significant role in this condition.

Ans. We recruited the patients with no identifiable causes of hypertension and randomized them so that the underlying diseases were comparable. Furthermore, most of the patients were middle-aged and elderly patients, with a mean age of 60 + 7.5 (range 26-70) years old in intervention group and 60.2 + 7.5 (range 36-70) in control group, which is an uncommon age group for primary aldosteronism.

4. The author provided detailed information about the randomization method and process. Line 207,“240 participants were enrolled from 6 clusters/group”, 12 clusters might be an accurate representation according to my understanding.

Ans. Participants were recruited from twelve clusters (villages) in six healthcare centers. (line 122) A total of 6 clusters were randomly allocated to the intervention group and another 6 clusters to the control group. (Line 129-131)

5.In the discussion section, the author noted that there was no adjustment of medication in patients. It is puzzling that the blood pressure in the control group still showed a significant decline.

Ans. We observed the trend in reduction of blood pressure and 24-hour urinary sodium excretion in the control group during follow-up clinic visit, indicating that standard health education was properly implemented in our study and may contribute to improvement in blood pressure control. (Line 297-300)

6.As mentioned before, the presentation of p-value should not minus the symbol = e.g. P 0.004.

Ans. We made change of P value as suggested.

---

## [Decision Letter · Decision Letter 2]

26 Aug 2024

PONE-D-24-07382R2Community-based intervention for monitoring of salt intake in hypertensive patients: A cluster randomized controlled trialPLOS ONE

Dear Dr. Kantachuvesiri,

Thank you for submitting your manuscript to PLOS ONE. After careful consideration, we feel that it has merit but does not fully meet PLOS ONE’s publication criteria as it currently stands. Therefore, we invite you to submit a revised version of the manuscript that addresses the points raised during the review process.

We look forward to receiving your revised manuscript.

Kind regards,

Peng Gao, Ph.D.

Academic Editor

PLOS ONE

Journal Requirements:

Reviewers' comments:

Reviewer's Responses to Questions

**Comments to the Author**

1. If the authors have adequately addressed your comments raised in a previous round of review and you feel that this manuscript is now acceptable for publication, you may indicate that here to bypass the “Comments to the Author” section, enter your conflict of interest statement in the “Confidential to Editor” section, and submit your "Accept" recommendation.

Reviewer #3: All comments have been addressed

Reviewer #4: All comments have been addressed

2. Is the manuscript technically sound, and do the data support the conclusions?

Reviewer #3: Yes

Reviewer #4: Yes

3. Has the statistical analysis been performed appropriately and rigorously? 

Reviewer #3: Yes

Reviewer #4: Yes

4. Have the authors made all data underlying the findings in their manuscript fully available?

Reviewer #3: Yes

Reviewer #4: Yes

5. Is the manuscript presented in an intelligible fashion and written in standard English?

Reviewer #3: Yes

Reviewer #4: Yes

6. Review Comments to the Author

Reviewer #3: (No Response)

Reviewer #4: In this manuscript, the authors explored the effects of salt intake monitoring in urine sodium excretion and blood pressure. However, there are some concerns need to be addressed.

1. Sine the baseline body weight and BMI were statistically higher in interventional group, the changes of these characteristics after follow-up should be provided and whether the changes were correlated with improver BP should be explored.

2. The details of medicine should be provided, especially SGLT2i.

3. The results should be presented with brief titles.

7. PLOS authors have the option to publish the peer review history of their article (what does this mean?). If published, this will include your full peer review and any attached files.

Reviewer #3: No

Reviewer #4: No

---

## [Author Response · Author response to Decision Letter 2]

19 Sep 2024

We response to reviewers’ comments below.

Reviewer #4: In this manuscript, the authors explored the effects of salt intake monitoring in urine sodium excretion and blood pressure. However, there are some concerns need to be addressed.

1. Since the baseline body weight and BMI were statistically higher in interventional group, the changes of these characteristics after follow-up should be provided and whether the changes were correlated with improver BP should be explored.

Answer. Since the baseline body weight and BMI were statistically higher in the interventional group, we studied changes in these characteristics after 12 weeks of follow-up. There was no significant difference (P=0.09) in the change in BMI from baseline between the two groups (BMI decreased by 0.03 and 0.23 kg/m2 in the intervention and control groups, respectively). Therefore, the noticeable changes in blood pressure were not attributable to weight reduction. (Lines 190-194 in revised manuscript with track changes)

2. The details of medicine should be provided, especially SGLT2i.

Answer. We did not consider lifestyle changes (e.g. exercise) and baseline anti-hypertensive medications. This is due to the fact that in the rural primary healthcare setting where this study was done, there is a limited list of first-line medications for high blood pressure (thiazide diuretics, calcium channel blockers and angiotensin-converting enzyme inhibitors). Sodium-glucose cotransporters-2 inhibitors are not available in this setting. In addition, there were no reports of any symptoms related to hypertensive complications, nor any adjustment of medications in our patients (Discussion on limitation of the study, Lines 249-254.)

3. The results should be presented with brief titles.

Answer. We put brief titles in the results as suggested.

1. In the online submission form you indicate that your data is not available for proprietary reasons and have provided a contact point for accessing this data. Please note that your current contact point is a co-author on this manuscript. According to our Data Policy, the contact point must not be an author on the manuscript and must be an institutional contact, ideally not an individual. Please revise your data statement to a non-author institutional point of contact, such as a data access or ethics committee, and send this to us via return email. Please also include contact information for the third party organization, and please include the full citation of where the data can be found.

Answer We have added the all relevant data in the Supporting Information file.

---

## [Editor Report · Decision Letter 3]

27 Sep 2024

Community-based intervention for monitoring of salt intake in hypertensive patients: A cluster randomized controlled trial

PONE-D-24-07382R3

Dear Dr. Kantachuvesiri,

We’re pleased to inform you that your manuscript has been judged scientifically suitable for publication and will be formally accepted for publication once it meets all outstanding technical requirements.

Kind regards,

Peng Gao, Ph.D.

Academic Editor

PLOS ONE
---

## [Editor Report · Acceptance letter]

2 Oct 2024

PONE-D-24-07382R3 

PLOS ONE

Dear Dr. Kantachuvesiri, 

I'm pleased to inform you that your manuscript has been deemed suitable for publication in PLOS ONE. Congratulations! Your manuscript is now being handed over to our production team.

Kind regards, 

on behalf of

Professor Peng Gao 

Academic Editor

PLOS ONE